# Correlation between Polysomnographic Parameters and Tridimensional Changes in the Upper Airway of Obstructive Sleep Apnea Patients Treated with Mandibular Advancement Devices

**DOI:** 10.3390/jcm10225255

**Published:** 2021-11-11

**Authors:** Sara Camañes-Gonzalvo, Rocío Marco-Pitarch, Andrés Plaza-Espín, Javier Puertas-Cuesta, Rubén Agustín-Panadero, Antonio Fons-Font, Carla Fons-Badal, Marina García-Selva

**Affiliations:** 1Department of Stomatology, Faculty of Medicine and Dentistry, University of Valencia, 46010 Valencia, Spain; saracamanes@outlook.com (S.C.-G.); aplaza_@hotmail.com (A.P.-E.); ruben.agustin@uv.es (R.A.-P.); antonio.fons@uv.es (A.F.-F.); carlafonsbadal@gmail.com (C.F.-B.); marina.garcia@uv.es (M.G.-S.); 2Medical School of Medicine, Universidad Católica de Valencia, 46002 Valencia, Spain; fj.puertas@ucv.es

**Keywords:** mandibular advancement device, obstructive sleep apnea syndrome, airway obstruction, sleep disordered breathing, oral appliance, snoring

## Abstract

Background. The effectiveness of mandibular advancement devices has been solidly demonstrated in the past. They are considered a valid alternative treatment to continuous positive airway pressure for patients with obstructive sleep apnea. Nevertheless, the relationship between polysomnographic parameters and the increase in the volume of the upper airway in patients with obstructive sleep apnea syndrome has not been clearly established so far. This study aimed to determine the impact of these oral appliances upon the volume of the airway after the device titration phase and correlate it with the degree of mandibular advancement and the improvement of polysomnographic parameters. Methods. All patients were diagnosed by polysomnography and were treated with a customized, titratable mandibular advancement device. Three-dimensional volumetric measurements were performed using cone beam computed tomography. Results. The present study included 45 patients diagnosed with obstructive sleep apnea hypopnea syndrome (mild in 23 patients, moderate in 11 and severe in 11). Forty-four percent of the patients presented with an apnea hypopnea index <5/h at the end of treatment. The volume of the upper airway increased an average of 4.3 ± 5.9 cm^3^, this represents a percentage increase of 20.9%, which was significantly correlated with an apnea hypopnea index and a minimum oxygen saturation improvement. Conclusions. The mandibular advancement device used was found to be effective in improving polysomnographic parameters. Moreover, the oral appliance was able to significantly increase the tridimensional dimensions of the upper airway. Moreover, this finding was correlated with a reduction in the apnea hypopnea index (*p* = 0.007) and an increase on minimum oxygen saturation (*p* = 0.033).

## 1. Introduction

Sleep disordered breathing (SDB) comprises a series of clinical conditions including simple snoring and obstructive sleep apnea hypopnea syndrome (OSA). Regarding the most recent definition of OSA, it is considered when there is either the presence of an apnea-hypopnea index (AHI) ≥15/h, which is predominantly obstructive, or when there is the presence of an AHI ≥5/h accompanied by one or more of the following factors: excessive sleepiness during the day, non-restorative sleep, excessive tiredness and/or deterioration of the quality of life related to sleep, not justifiable for other reasons [1]. Simple snoring is defined as hoarse breathing sounds produced in the upper airway (UA) during sleep, without the presence of airway obstruction. Although it may cause social problems, snoring is not associated with daytime sleepiness and has no pathological consequences [2], in contrast with OSA.

The diagnosis of OSA requires an exhaustive exploration of a physician specialized in sleep breathing disorders, which includes certain screening tests such as the Mallampati score [3]. Moreover, the confirmation of clinical suspicion with a sleep test is required: standard PSG (level I), domiciliary PSG (level II) or ambulatory respiratory polygraphy (level III) [4]. To date, the AHI, corresponding to the number of apnea hypopnea events per hour of sleep, has been regarded as the main parameter for determining the severity of OSA, with the definition of three categories: mild (5 ≤ AHI < 15), moderate (15 ≤ AHI < 30) and severe (30 ≤ AHI).

However, according to the latest international consensus on obstructive sleep apnea, other parameters such as minimum oxygen saturation and the oxygen desaturation index (ODI), defined as the number of episodes of oxygen desaturation per hour of sleep, should be considered to determine the severity of OSA. On the other hand, the literature also uses the respiratory disturbance index (RDI), defined as the number of respiratory events, including apneas, hypopneas and respiratory effort-related arousals (RERA) per hour of sleep, instead of IAH [4]. However, in the most recent consensus, it was concluded that respiratory effort-related arousals should not be distinguished from hypopneas and, therefore, it is better to use the AHI parameter [5].

The literature classically describes a prevalence of OSA of approximately 9–38% in the general adult population, with a greater presence among men. However, the incidence of OSA changes with age in the elderly population, with 88% of men aged 65 to 69 years having five or more events per hour, while a higher percentage of up to 90% is seen in men aged 70 to 85 years [6,7].

Nevertheless, a lack of awareness of the condition among the general population and on the part of clinicians implies that 80–90% of all individuals with SDB have not been correctly diagnosed [8,9].

With regard to treatment, continuous positive airway pressure (CPAP) is the most widely used and effective option [10,11,12]. However, due to initial patient reluctance and middle- to long-term dropouts, approximately 30% of all individuals with a prescription of CPAP are not adequately treated, with the consequent persistence of the clinical consequences of the disorder. This situation evidences the need for alternatives to CPAP in the management of SDB [12].

Taking CPAP’s limitations into account, the surgical approach must also be considered. Palatal and oropharyngeal surgeries can be offered precisely based on clinical findings and endoscopy using induced sedation, but skeletal surgery, especially bimaxillary advances, may also be indicated as an initial surgical treatment of OSA in severe OSA patients with AHI > 65 and/or concentric collapse on endoscopy through induced sedation and/or severe dentofacial alterations [5].

Moreover, several authors have demonstrated the efficacy of barbed techniques as such expansion sphincter pharyngoplasty (ESP), barbed reposition pharyngoplasty (BRP), and lateral pharyngoplasty approaches on muscle tension and muscle fiber tearing. However, selection plays an important role in patient management and surgical outcome on obstructive apneas [13,14].

Since 2015, the American Academy of Sleep Medicine (AASM) and the American Academy of Dental Sleep Medicine (AADSM) have recommended the prescription of mandibular advancement devices (MADs) performed and titrated by qualified dentists experienced in the treatment of patients with snoring, mild or moderate OSA or severe OSA who reject or do not tolerate CPAP [4].

There is solid evidence for the efficacy of MADs. The use of such devices is associated with a substantial decrease (over 50%) in the initial AHI, and an AHI < 10 is achieved in 30–83% of all patients [4,15,16,17,18]. In addition to reducing the AHI, these devices improve other polysomnographic parameters including oxygen desaturation, sleep architecture and the microarousal index [16,19].

Nevertheless, although the efficacy of MAD has been demonstrated, the relationship between polysomnographic parameters, such as the reduction in the AHI after MAD placement or the increase in volume of the upper airway in patients with SDB, has not been clearly established to date [9].

Most studies have used lateral cephalometric radiographs to analyze dimensional changes of the UA. However, although such procedures are generally reliable in assessing skeletal structures, they are unable to afford precise evaluations of volumetric changes [20,21,22]. One of the most widely available tools capable of reproducing volumetric dimensions and calculating it in an affordable manner for the dentist is cone beam computer tomography (CBCT). Although CBCT is not the best method to evaluate soft tissues, it offers the clinician a three-dimensional exploration accompanied by a low radiation dose [23].

The present study was carried out to determine the impact of mandibular advancement devices upon the volume of the airway after the device titration phase. As secondary objectives, an analysis was conducted of the correlation between the degree of mandibular advancement and the volumetric changes in the airway, and the correlation between the improvement of respiratory variables (AHI, minimum oxygen saturation (SaO_2_Min) and arousal index) and the volumetric increase in the UA.

## 2. Materials and Methods

### 2.1. Study Design

A prospective longitudinal panel study was carried out at the University of Valencia (Valencia, Spain). It is considered a longitudinal panel study, since the same population was studied in different periods of time. All patients gave informed consent for participation in the study, which was approved by the local research ethics committee (H17052010). The interpretation of the PSG was performed according to the definitions included in the guidelines of the American Academy of Sleep Medicine (AASM) [24].

### 2.2. Study Sample Identification and Selection

All patients were referred to the Prosthodontics and Occlusion Unit (University of Valencia) from the Ear, Nose and Throat Department (Valencia University Clinic Hospital, Valencia) and the Sleep Unit of the Neurophysiology Department (La Ribera University Hospital, Alzira, Valencia). After the pertinent diagnostic and exploratory tests, the specialist in SDB referred those patients diagnosed to the Prosthodontics and Occlusion Unit. All patients underwent a comprehensive examination by a specialized otolaryngologist in respiratory sleep disorders. The examination included a sinonasal evaluation, a nasal endoscopic examination, and an evaluation of the oral and oropharyngeal cavity (uvula size, oropharyngeal morphology, soft palate, relationship between palatoglossal and palatopharyngeal muscles and uvula insertion, and tongue using the modified Mallampati scale and the Friedman tonsil examination), in addition to a nasofibrolaryngoscopic evaluation and cervical-facial morphology [25]. In addition, those patients who were candidates for surgery but ultimately candidates for oral devices underwent sleep endoscopy.

Patients older than 18 years old diagnosed with mild or moderate OSA patients or severe OSA who failed or refused treatment attempts with either CPAP or upper airway surgery were included in the study.

Patients with less than 10 teeth per dental arch, acute temporomandibular joint disorders, active periodontal disease or a protrusive capacity of less than 5 mm, and pregnant women were excluded from the study.

### 2.3. Treatment Procedures

In order to perform the volumetric measurements, each patient underwent two CBCT scans separated by an approximate time frame of three months: one at the start of the study, without the mandibular advancement device, and another at the end of the MAD titration phase, with the device in the mouth of the patient. All the studies were performed in the Radiology Unit (University of Valencia) by using the Master 3D conical beam system (90 kVp and 4 mA in 30 s, 300 µm voxel). With the purpose of standardizing all the projections, the patient was placed in the sitting position with the Frankfort plane parallel to the floor; the vertical laser beam was made to coincide with the midsagittal plane of the patient, and the horizontal laser beam was made to coincide with the lower margin of the mandibular lip. The patient was instructed to breathe through their nose and was not allowed to swallow during the scan, keeping the teeth in occlusion.

The MAD used in the present study (DAM^®^, Aditas, Asturias, Spain) is a customized, titratable two-piece device consisting of two acrylic resin splints jointed by a screwed plate in the incisor zone (Figure 1). The exchangeable stepped plate allows variable advancement of the mandibular splint. The device is supplied with seven plates, allowing sequential millimetric mandibular advances. This device allows wide and controlled mobility, practically natural, but limited.

The initial advancement recorded for every patient was 50% of the maximal protrusion and it was measured using the George gauge (Great Lakes Orthodontics Ltd., Bay City, MI, USA) [5,26]. Patients were recalled every 2 weeks, and according to the responses given by the patient in the domiciliary questionnaire about the subjective changes noticed and the presence of secondary effects, the clinician titrated the MAD, until the appliance was considered regulated. Once the MAD was adjusted, the PSG was repeated with the appliance in the mouth, at least 3–6 months after definitive MAD titration. This is the period of time in which the device produces the maximum benefit to the patient with the minimal appearance of side effects. This period is similar to the general adjustment time period [5].

### 2.4. Study Variables and Data Collection

The CBCT data with and without the DAM^®^ were exported to the Invivo6 software (KaVo Kerr, Biberach, Germany) in DICOM format (479 images each). In order to maximally standardize the study methodology, the three-dimensional reconstructions were oriented to ensure that the horizontal Frankfort plane was parallel to the axial plane. In the axial view, a vertical line was traced through the posterior nasal spine (PNS), centering the sagittal view on the facial midline of the patient (Figure 2a). The reference employed to standardize the coronal plane was the line between the infraorbital points, named the infraorbital line. The airway tool of the application was used to define the measurement limits between the nasopharynx and the hypopharynx. Specifically, the volume between the ceiling of the cavum and the termination of the cricoid cartilage was analyzed (Figure 2b). The measurements of the CBCT images were obtained by a calibrated examiner who was blind to other study data, such as anthropometric elements, physical examination and polysomnography. The intra- and inter-examiner reproducibility of CBCT measurements of airway volume were investigated by repeating measurements of the variables 2 weeks apart by two experienced oral radiologists.

The efficacy of the device was assessed in OSA patients by calculating the AHI, MinSaO_2_ and arousal index post-titration parameters. Nevertheless, the heterogeneity encountered in the literature regarding the definition of a successful post-titration AHI is quite conflictive [27]. In the present study, complete response was defined as an AHI < 5 after DAM^®^ titration.

### 2.5. Statistical Analysis

The SPSS 15.0 (IBM SPSS statistics, Chicago, IL, USA) software was used for statistical analysis. A descriptive statistical analysis was performed, with calculation of the mean and standard deviation (SD) and range. Normal data distribution was assessed by means of the Kolmogorov–Smirnov test. The inferential analysis was based on the Student *t*-test for repeated measures to contrast the variation of AHI over time, with application of the Wilcoxon test in the case of data without a normal distribution, and the Spearman nonlinear correlation coefficient to explore associations between changes in the different parameters.

The Student *t*-test was used to evaluate possible significant differences between the baseline and definitive values. The test afforded a power of 93.3% in identifying a variation as being significant, assuming a 95% confidence level. Statistical significance was considered for *p* < 0.05. The- intra and inter-examiner reproducibility of CBCT measurements was evaluated using the intraclass correlation coefficient (ICC) and the Bland and Altman agreement tests, with their respective confidence intervals set at 95%.

The sample size was calculated using two independent means based on the AHI parameter. Accepting an alpha risk of 0.05 and a beta risk of 0.3 in a bilateral contrast, 50 subjects were required to detect a difference equal to or greater than 2.2 units.

## 3. Results

### 3.1. Descriptive Analysis of the Sample

Patient data were collected consecutively over a period of three years. Sufficient data for analysis were obtained from 47 patients. Two patients were unable to sleep with the device (Figure 3).

The final sample included 45 patients diagnosed with OSA (mild in 23 patients, moderate in 11 and severe in 11). The study sample comprised 24 men and 21 women, with a mean age of 54 years, and a mean body mass index (BMI) of 26 kg/m^2^. The baseline demographic and clinical characteristics are shown in Table 1.

### 3.2. Impact of Mandibular Advancement upon the Respiratory and Neurophysiological Parameters

The mean protrusion at which the MAD was titrated was 72.6 ± 14.9% of the maximum protrusion of the patient. The MAD titration time from the delivery of the device with the initial advancement to the obtainment of maximum benefit with the definitive advancement was 5.6 ± 2.7 months on average.

On the other hand, AHI decreased from 21.9 ± 16.6 before placement of the device to 9.1 ± 11.2 after MAD titration (*p* < 0.001). MinSaO_2_ increased from 84.6 ± 6.9% before the placement of the device to 87.9 ± 4.9% after MAD titration (*p* < 0.001). The arousal index decreased from 18.1 ± 14.7 before the placement of the device to 15.8 ± 16.0 after MAD titration (*p* = 0.255). In Table 2, OSA patients are classified according to the AHI severity at baseline and to its evolution after MAD placement. A patient was considered a complete responder when their AHI was < 5 after the appliance titration. According to this, the number of complete responders after MAD placement was 22. Mild patients reduced by 4.2 events/h on average, moderate patients reduced by 17 events/h on average, and severe patients reduced by 26.6 events/h. The relationship is clear—those who are worse off are those who improve the most (Table 2). Significant differences between the three groups are found (*p* < 0.001) (Figure 4).

### 3.3. Impact of Mandibular Advancement upon the Structure of the Airway

After MAD placement and titration, the volume of the upper airway was seen to increase from 20.6 ± 6.8 to 24.9 ± 8.8 cm^3^ (*p* < 0.001), with a mean total increment of 4.3 ± 5.9 cm^3^. This represents a percentage increase of 20.9% (Table 3).

Figure 5 shows the increase in volume of the upper airway in the lateral, anterior and posterior views corresponding to one of the patients.

### 3.4. Correlation between the Morphological Changes in the Airway and the Polysomnographic Parameters

An analysis was conducted of the correlations between different parameters, with a view to improving our understanding of the mechanism of action of the mandibular advancement devices. Specifically, the correlation between the degree of mandibular advancement and the changes in volume of the UA and the correlation between improvement of AHI and the volumetric changes of the airway was assessed.

The analysis of the relationship between the degree of mandibular advancement and the changes in volume of the UA yielded a null correlation (r = −0.02; *p* = 0.922), meaning that gradual advancement of the MAD did not influence the increase in volume of the UA (Figure 6).

In addition, improvement in AHI was correlated with changes in airway volume. It was observed that as airway volume increased, there was an improvement in AHI, but this was only apparent from the volume increase of approximately 5 cm^3^. The correlation was significant (r = −0.40; *p* = 0.007) (Figure 7).

When correlating the same parameters according to the severity of baseline AHI, this relationship was only statistically significant among severe patients. In other words, only in severe patients was it possible to demonstrate that a greater increase in the airway volume was related to greater reductions in AHI (r = −0.82; *p* = 0.002) (Figure 8).

On the other hand, it was observed that a greater increase in volume was related to a greater increase in minimum oxygen saturation (r = 0.32; *p* = 0.033) (Figure 9).

However, the correlation between a greater increase in the upper airway volume and the arousal index reduction was not statistically significant (r = −0.25; *p* = 0.198) (Figure 10).

Excellent reliability was observed for assessed variables (total volume). Regarding the intra-examiner errors of Researcher 1, high reliability was observed for the variables (ICC ranging from 0.98 to 0.99), with very narrow confidence intervals, thereby showing excellent agreement for these measurements. Similar results were observed for the intra-examiner errors of Researcher 2, with ICC values ranging from 0.93 to 0.99. The inter-examiner evaluation also showed high reliability, with ICC ranging from 0.88 to 0.98 (Table 4).

## 4. Discussion

The study sample comprised a typical population of patients diagnosed with OSA and snoring disorders by physicians specialized in SDB. It was considered to be a representative sample of 45 individuals extrapolatable to the general population.

The results of this prospective clinical study demonstrated the efficacy of MAD, with statistically significant improvement of AHI after placement of the device. The definition of treatment success has not been standardized to date in the literature on the efficacy of MAD [27]. Some authors define success as the achievement of AHI < 10, while others propose AHI < 15 or a 50% decrease in initial AHI. More rigorous studies even define treatment success as AHI < 5 [28,29,30]. In the present study, 44% of the patients completed treatment with less than five events/hour. Similar findings have been published by other authors [31,32].

Moreover, those patients who had a more severe AHI at baseline responded better. This is an interesting result, because the literature defines an inverse relationship [33]. However, in this study, severe OSA patients have experienced a significant AHI reduction. Notably, all severe patients were previously treated with CPAP and did not tolerate it.

Several studies show that, for the phenotype characterized in sleep endoscopy with retrolingual collapse from hypertrophic lymphatic tonsil compared to the phenotype of the muscular tongue, excellent control of AHI has been shown in the literature when more than 10 cc of soft tissue is removed through the robotic TORS approach. This technique allows for a minimally invasive approach with no swallowing disorders, bleeding, or pain [34,35].

Regarding MinSaO_2,_ a significant improvement was observed in this investigation. These results are similar to those obtained in other studies that also used a custom-made device [36,37].

One of the pathological consequences of OSA is sleep deprivation due to arousals. In order to promote a more restful sleep, a reduction in the arousal index is expected through treatment with MAD [38]. There are multiple studies that evaluate the changes produced in sleep fragmentation, and that verify that there is a reduction in the arousal index, by using a custom-made and adjustable MAD [38,39,40]. However, this reduction was not statistically significant in this study. These results are similar to those obtained by other authors [11,41]. These discrepancies could be due to the use of a non-adjustable device in the case of one of these studies, or due to the use of a type II home monitoring system to evaluate effectiveness, in the other study. In this study, there was a non-significant reduction in the arousal index, so a higher sample is probably needed to re-evaluate this parameter.

A CBCT was performed to evaluate the three-dimensional changes to the airway. Although this tool has certain limitations, since it does not allow a clear visualization of the soft tissues, it offers a series of advantages versus two dimensional radiographs [19].

In the present study, by using the Invivo6 application, a mean volume increase of 4.3 ± 5.9 cm^3^ in the global sample after titration of the device was recorded. Similar findings have been published by other authors [42,43,44]. Chan et al., by using the 3D Amira image analyzing application, also recorded a significant increase in total volume (1.6 cm^3^ on average) [45]. According to Haskell, this increase in volume implies a better air flow in the UA, thereby reducing the patient symptoms [46].

The point of the UA with the smallest cross section is where its lumen is theoretically most limited during the waking state; as such, it represents an air flow limiting factor and may represent a point of obstruction during sleep in patients with OSA. The outcomes of the different treatment modalities in these patients are conditioned by an increase or decrease at this point in the UA with the smallest cross section; an increase in the latter therefore constitutes a definitive indicator of improved air flow in the UA [46]. In our study, MAD effectively increased the cross section at the most compromised point of the UA (Figure 5).

It has been shown that mandibular advancement results in an enlargement of a narrow UA [11,41,42]. However, it is not clear whether there is a linear relationship between the magnitude of mandibular advancement and the increase in airway volume. Tsuiki et al. were pioneers in the analysis of this relationship and reported greater anteroposterior changes in the velopharynx with increasing advancement of the MAD (33%, 67% and 100% of maximum protrusion of the device) [47]. Likewise, Choi et al. observed a decrease in airway collapsibility according to the magnitude of mandibular advancement (1/3, 2/3 and maximum mandibular advancement) [48]. In a later study, Gao et al., by using a customized titratable device, recorded a correlation between different degrees of mandibular advancement (0%, 50%, 75%, 100%) and the volume of the airway [49].

However, in this study, no correlation was observed between mandibular advancement and an increased caliber of the UA. This is consistent with the observations of Zhao et al., who also used a customized device and postulated that this lack of a correlation may be due to inter individual differences among patients [42]. L’Estrange et al. demonstrated the existence of significant inter-individual variations in the dimensional changes of the UA, which appear to be attributable to differences in facial anatomy [50]. An unfavorable facial structure is believed to play a relevant role in the pathogenesis of SDB. This in turn could be accompanied by an altered response of the pharyngeal dilatory muscles [51].

Thus, the lack of a correlation between mandibular advancement and the volume of the UA observed in our patients suggests that mandibular advancement is not the only factor influencing the volume of the airway, and that the mandibular morphological features of the patient may also play a role [52].

On the other hand, the possible correlation between the increase in volume of the airway and improvement of AHI was evaluated. In this regard, increased volume was associated with a decrease in AHI, but this was only apparent from a volume increase of approximately 5 cm^3^. This is consistent with the results published by Liang et al., who demonstrated, with the use of the Müller maneuver, that the increase in the upper airway at the level of the velopharynx was correlated with the reduction in the AHI [53]. Shete et. al., by using a modified twin block device, likewise recorded a negative correlation, though without reaching statistical significance [44]. However, other authors did not find a significant association between those parameters [54,55]. Sutherland et al. attributed it to the variability of OSA from one night to other, and to the different body positions of the patient during sleep [54].

In the present study, when correlating the same parameters according to the severity of baseline AHI, this relationship was only statistically significant among severe patients. In other words, only in severe patients was it possible to demonstrate that a greater increase in the airway was related to greater reductions in AHI. This is consistent with the results published by Vos W et al. [56].

Moreover, the possible correlation between the increase in the upper airway volume and an improvement in minimum oxygen saturation was evaluated. In this regard, increased volume was associated with an improvement in the MinSaO_2._ However, a statistically significant correlation between the increase in the volume and the reduction in the arousal index was not found. These results are consistent with those obtained by Marco et al. using the same device [53].

An improvement in AHI and a reduction in MinSaO_2_ reduction are not only due to an increase in volume of the UA, and there are some aspects of the mechanism of action of these devices that are still not completely clear. In general, the literature indicates that the advancement produced by the device results in an increase in the volume of the UA, favoring air flow and acting upon the anatomical factor [39]. However, the fact that patients with SDB have an UA that is more collapsible than in healthy subjects is attributable to the coexistence of anatomical and neuromuscular factors [57,58,59].

Therefore, it would be interesting to carry out studies with larger samples to explore the behavior of the UA in these patients and correlate it with the different respiratory and neurophysiological parameters.

### Limitations

A limitation of the present study is the fact that the CBCT scans were obtained with the patients in the waking state and seated. In patients with SDB, while the patient remains awake, the activation of the pharyngeal muscles is successfully performed, hence preventing its collapse. Therefore, the pharyngeal sizes determined in our study might not completely correspond to the UA dimensions and behavior during sleep, which is characterized by the closure or narrowing of the pharynx secondary to a decrease in muscle tone [60,61].

On the other hand, a nonrandomized study design was adopted, with the purpose of analyzing the effects of MAD in patients in need of treatment referred by an expert in SDB. Regarding the severe medical consequences associated with SDB, the possibility of including a placebo group seemed unethical, and was therefore discarded.

## 5. Conclusions

The mandibular advancement device used in the present study has been shown to be effective in patients with OSA and simple snoring, reducing AHI in the cases of OSA and improving the patient symptoms.

Considering the mechanism of action of these advancement devices, the MAD is able to increase the volume of the upper airway, improving air flow and thus restoring airway permeability. Moreover, there is a correlation between the volume increase and the AHI reduction and MinSaO_2_ augmentation. Regarding the AHI index, this correlation was apparent from the volume increase of approximately 5 cm^3^, and was only statistically significant among severe patients.

However, revealing the impact of the mandibular advancement in the upper airway and the muscles inserted in it should acquire a high priority for future studies. It is increasingly acknowledged that SDB is characterized by the heterogeneity of its underlying risk factors, clinical presentation, pathophysiology, comorbidities and response to treatment. A detailed phenotypic analysis of these patients is therefore required in order to afford individualized treatment and ensure its effectiveness. 

## Figures and Tables

**Figure 1 jcm-10-05255-f001:**
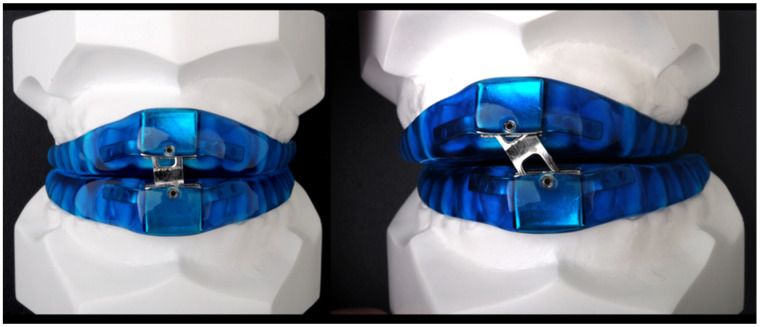
DAM^®^: a customized, titratable two-piece mandibular advancement device.

**Figure 2 jcm-10-05255-f002:**
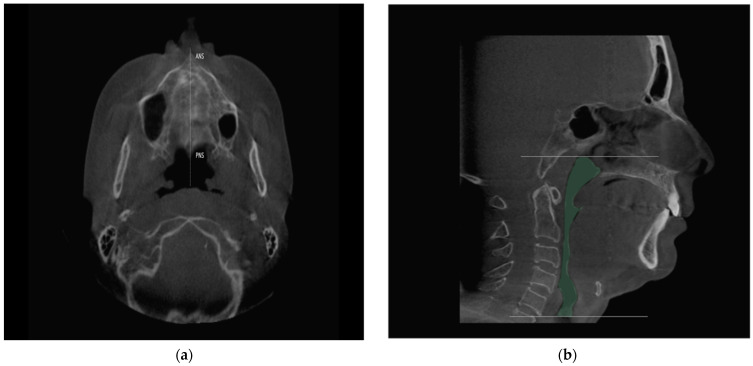
(**a**) Orientation of the three-dimensional reconstruction for the volumetric measurement, from the anterior nasal spine to the posterior nasal spine; (**b**) determination of the limits of the upper airway for the volumetric measurement. ANS: Anterior nasal spine; PNS: Posterior nasal spine.

**Figure 3 jcm-10-05255-f003:**
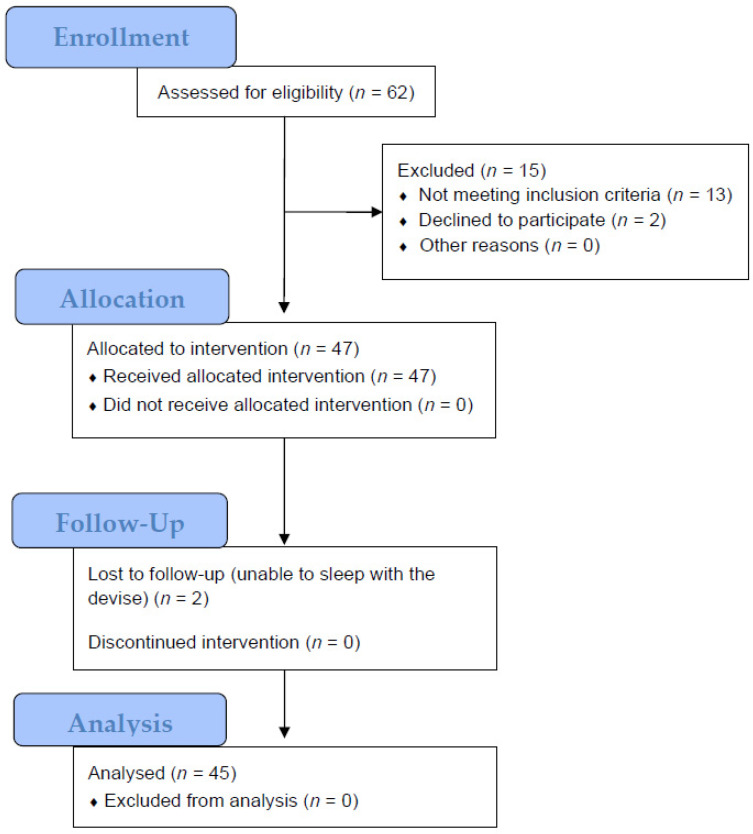
Study process adapted from CONSORT 2010 flow diagram.

**Figure 4 jcm-10-05255-f004:**
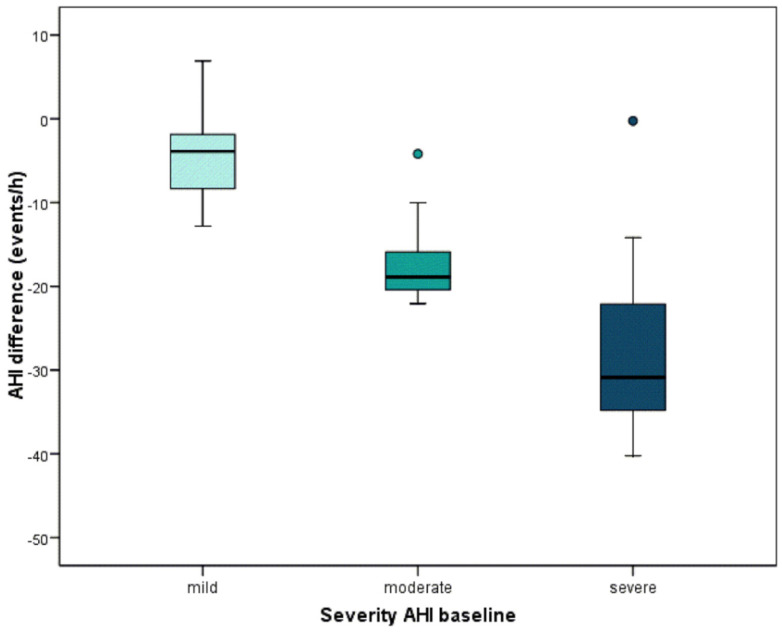
Significant differences between the three groups are found (*p* < 0.001) through Kruskal–Wallis test. Those patients who are worse off are those who improve the most.

**Figure 5 jcm-10-05255-f005:**
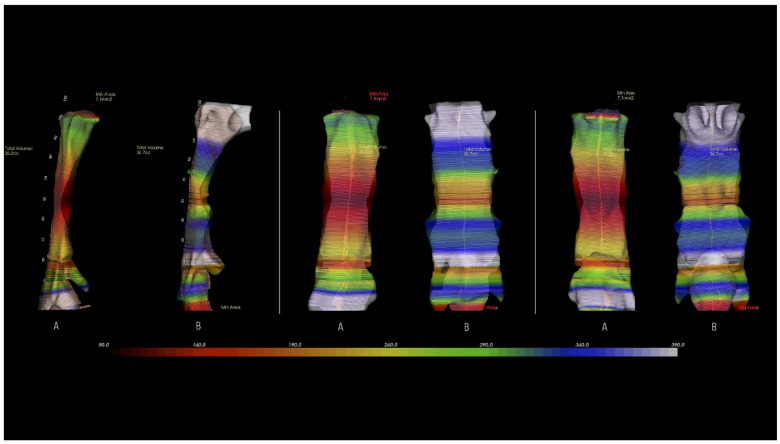
Three-dimensional rendering of the upper airway volume in lateral, anterior, and posterior views: pretreatment (**A**) and posttreatment (**B**) with the mandibular advancement device.

**Figure 6 jcm-10-05255-f006:**
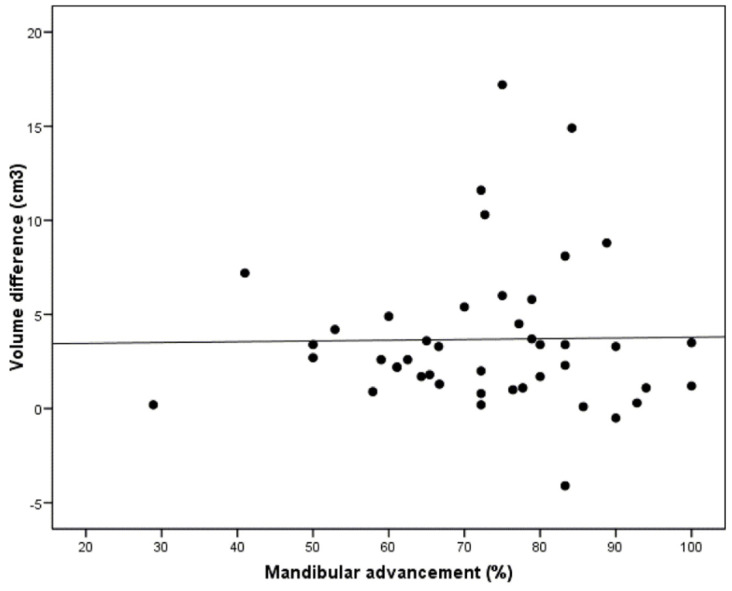
Correlation between the degree of mandibular advancement and the volumetric changes of the upper airway (*p* = 0.922).

**Figure 7 jcm-10-05255-f007:**
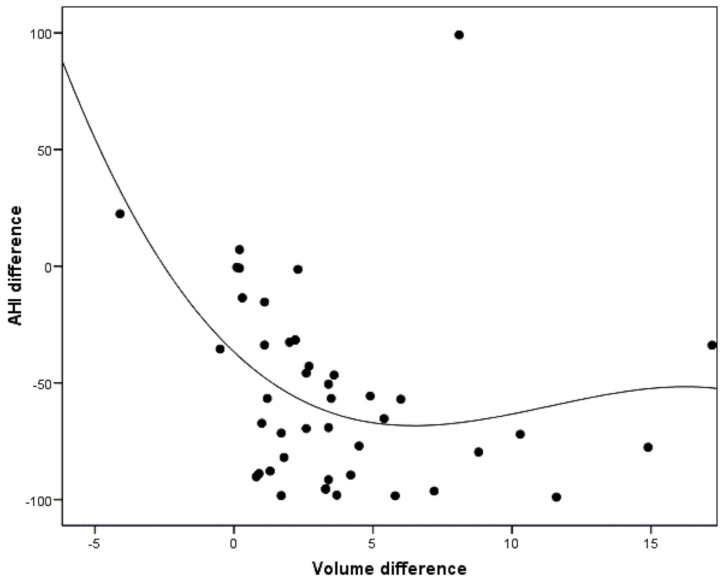
Correlation between the changes in AHI after MAD titration and the volumetric changes in the upper airway (*p* = 0.007).

**Figure 8 jcm-10-05255-f008:**
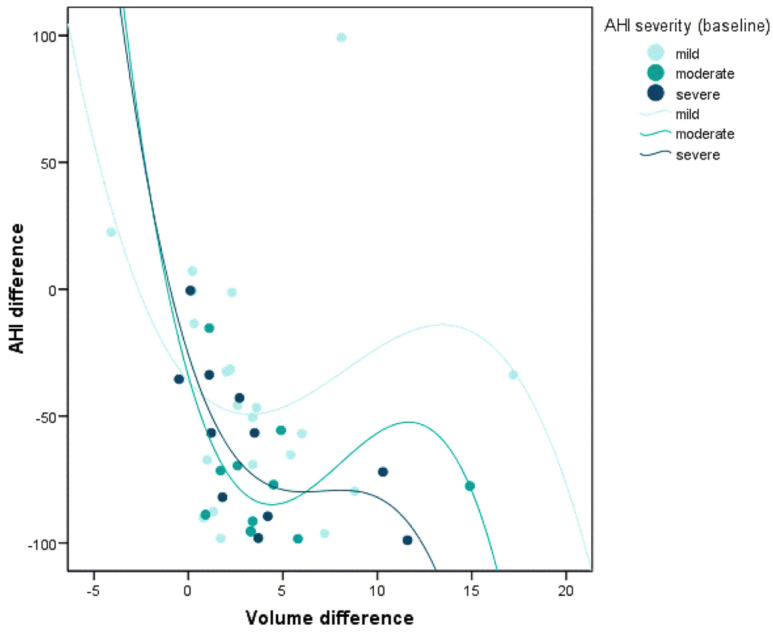
Correlation between the changes in AHI after MAD titration and the volumetric changes of the upper airway according to the OSA severity (*p* = 0.002).

**Figure 9 jcm-10-05255-f009:**
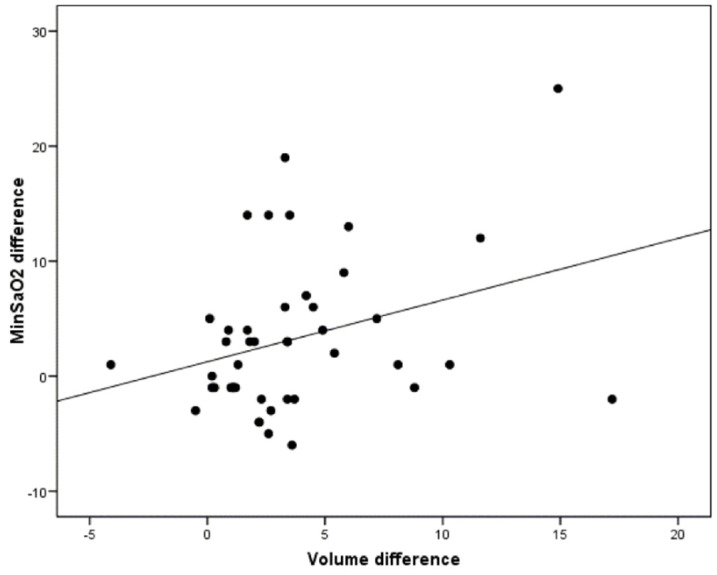
Correlation between the changes in minimum oxygen saturation after MAD titration and the volumetric changes of the upper airway according to the OSA severity (*p* = 0.033).

**Figure 10 jcm-10-05255-f010:**
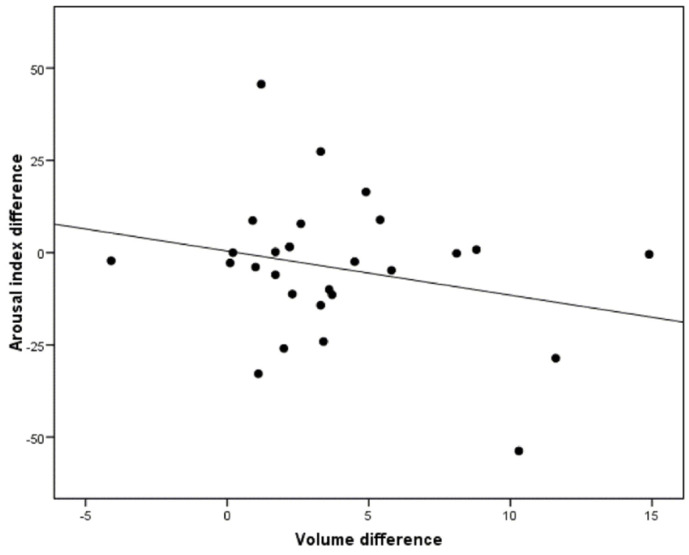
Correlation between the changes in arousal index after MAD titration and the volumetric changes of the upper airway according to the OSA severity (*p* = 0.198).

**Table 1 jcm-10-05255-t001:** Demographic characteristics of the 50 patients treated with the mandibular advancement device (MAD).

Demographic Characteristics	Baseline Measure
Gender, *n* (%)	
Men	24 (53.3)
Women	21 (46.7)
Age, years, mean (SD) ^1^	54.1 (10.2)
BMI, kg/m^2^, mean (SD) ^2^	25.6 (2.7)
AHI, events/r, mean (SD) ^3^	21.9 (16.6)
MinSaO_2_, mean (SD) ^4^	84.6 (6.9)
Arousal index	18.1 (14.7)
OSA severity category, *n* (%) ^5^	
Mild (5 ≤ AHI <15)	23 (51.1)
Moderate (15 ≤ AHI <30)	11 (24.4)
Severe (30 ≤ AHI)	11 (24.4)

^1^ SD: standard deviation; ^2^ BMI: body mass index; ^3^ AHI: apnea-hypopnea index; ^4^ MinSaO_2_: minimum oxygen saturation; ^5^ OSA: obstructive sleep apnea syndrome.

**Table 2 jcm-10-05255-t002:** AHI changes once the MAD was titrated according to AHI severity.

	AHI Severity			
	Total	Mild	Moderate	Severe
N	45	23	11	11
mean (SD)	−12.8 (11.8)	−4.2 (4.5)	−17.0 (5.5)	−26.6 (11.9)
minimum	−40.2	−12.8	−22.1	−40.2
maximum	6.9	6.9	−4.2	−0.3
percentile 25	−20.9	−8.5	−21.0	−36.8
median	−9.7	−3.9	−18.9	−30.9
percentile 75	−3.9	−1.8	−14.0	−20.9

**Table 3 jcm-10-05255-t003:** Changes in total airway volume before and after treatment with the mandibular advancement device (MAD).

	Baseline	After MAD Tritation	*p*-Value ^1^
Total airway volume (cm^3^), mean ± SD	20.6 ± 6.8	24.9 ± 8.8	*p* < 0.001

^1^*p* value for the paired *t*-test.

**Table 4 jcm-10-05255-t004:** Intra- (Researcher 1 and Researcher 2) and inter-examiner (Researcher 1 vs. Researcher 2) errors for total volume: mean, standard deviation (SD), Bland and Altman agreement and intraclass correlation coefficient (ICC).

Researcher 1 1st Measurement	Researcher 1 2nd Measurement	Bland and Altman			ICC
Mean ± SD	Mean ± SD	Mean difference	SD	95% confidence interval
15.5 ± 5.9	15.2 ± 5.7	10.3	360.02	−695.32; 715.96	0.99
Researcher 2 1st measurement	Researcher 2 2nd measurement	Bland and Altman			ICC
Mean ± SD	Mean ± SD	Mean difference	SD	95% confidence interval	
15.8 ± 6.8	15.5 ± 5.1	74.3	637.57	−1175.36; 1323.94	0.99
Researcher 1 1st measurement	Researcher 2 1st measurement	Bland and Altman			ICC
Mean ± SD	Mean ± SD	Mean difference	SD	95% confidence interval	
15.5 ± 5.9	15.8 ± 6.8	−306.28	1573.3	−3390.36; 2777.53	0.95

## Data Availability

The data presented in this study are available on request from the corresponding author.

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
