# Peer review of "Correlation between Polysomnographic Parameters and Tridimensional Changes in the Upper Airway of Obstructive Sleep Apnea Patients Treated with Mandibular Advancement Devices"

_jcm, 2021, doi:10.3390/jcm10225255_

Round 1
Reviewer 1 Report
Minor corrections:
Abstract
- line 23, please define OSA severity
- line 25, please report volume percentage increase
- line 30 please report statistical significance for both
Introduction
- Line 36, please define OSAS according to the latest AASM criteria and please cite doi: 10.5664/jcsm.6576.
- line 44, please specify which PSG
- line 48, literature data confirm also the validity of RDI and ODI, and LOS. Please briefly describe the parameters
- line 51, the incidence of OSA changes with age in the elderly population, with 88% of men aged 65 to 69 years had 5 or more events per hour, while higher percentage up to 90% in men aged 70 to 85 years please cite doi:10.20452/pamw.15283.
- line 56, please report CPAP references in regard.
- line 60, Several authors have demonstrated the efficacy of barbed techniques as expansion sphincter pharyngoplasty (ESP), barbed reposition pharyngoplasty (BRP), and LT approaches on muscle tension and muscle fiber tearing. However, selection plays an important role in patient management and surgical outcome on obstructive apneas. please cite doi:10.1007/s00405-020-05883-2 and doi:10.1111/coa.13027
- which type of longitudinal study was performed? please specify
- line 103 patients performed ENT consulting? which examination was performed? Muller? Sleep endoscopy?
Statistical analysis 170
- which software was used for statistical analysis?
Table 1 add baseline (mean and sd) suppose to the legend
Results
- line 215 include percentage
Figure 5 is very interesting
Figure 6 and 7 and 8, 9, 10 please add p-value in the figure
Discussion
line 286, For the phenotype characterized at sleep endoscopy with retrolingual collapse from hypertrophic lymphatic tonsil compared to the phenotype of the muscular tongue, excellent control of AHI has been shown in the literature when more than 10 cc of soft tissue is removed through the robotic TORS approach. This technique allows for a minimally invasive approach with no swallowing disorders, bleeding, or pain. Please cite please cite doi:10.7717/peerj.7812 and doi:10.1002/rcs.2106
Reviewer 2 Report
The article shows the results of the author's research about volumetric changes of the airway after mad's insertion by means of a double cbct: one before treatment and 1 during the treatment. This is not a new topic in literature and the results are similar to the already published results. It's questionable to double scan a patient with a cone beam without any diagnostic purpose. The study is well designed and conducted. The major result is that, as we already clinically know, the volumetric increase of the airway is not proportional to the protrusion we offer to the patient. The second major result is that we must offer, at least, a 5 cm3 increase to have an improvement of AHI. As we already know cbct exams are taken with a seated and awake patients, very far from the ideal situation.
Reviewer 3 Report
Thank you for the opportunity to revise this manuscript.
Overall the manuscript is well written and of great interest. However, there are certain points that need to be clarified before publication:
1. In the Introduction, reference should be made not only to CPAP treatment but also to surgical treatment as an alyernative treatment in patients with OSA. Bimaxillary advancement has been shown to be highly effective as a treatment for severe OSA and it should be stated.
2. Line 127 states that MAD activation is started at 50%. It should be explained why activation is started at 50%. Is it random? If not, it should be justified with bibliographic references.
2. Line 132 states that a control PSG is performed 3-6 months after MAD titration. The study protocol should state whether the PSG is performed at 3 or 6 months. Is the timing of the control PSG randomised?
3. The study does not mention loss of efficacy of MAD, when it is clearly demonstrated that MAD loses efficacy over time. This should be explained in the discussion and in the limitation of the study.
Author Response
Please see the attachment.

This manuscript is a resubmission of an earlier submission. The following is a list of the peer review reports and author responses from that submission.
Round 1
Reviewer 1 Report
The authors are commended for their work on this project, however several items need clarification in order to improve the impact of this work:
Abstract: The first line of the conclusion should be more specific and state "in half of the patients" As only 50% of the patients had AHI reduced below 5 it is misleading to state generally that the treatment is effective. Also as the correlation between airway volume increase and AHI reduction was not strong, the conclusion should also state this.
Introduction:
Line 75 - reference 16 is not appropriate for the sentence as it does not directly assess CBCT as a "precise and useful diagnostic tool" for the airway.
It would be helpful if the authors included information on the reliability of CBCT assessment of the airway. See:
Obelenis Ryan DP, Bianchi J, Ignácio J, Wolford LM, Gonçalves JR. Cone-beam computed tomography airway measurements: Can we trust them? Am J Orthod Dentofacial Orthop. 2019 Jul;156(1):53-60. doi: 10.1016/j.ajodo.2018.07.024. PMID: 31256838.
Zimmerman JN, Vora SR, Pliska BT. Reliability of upper airway assessment using CBCT. Eur J Orthod. 2019 Jan 23;41(1):101-108. doi: 10.1093/ejo/cjy058. PMID: 30184085.
Methods:
Line 86 - authors should clarify that the scoring of PSGs followed the AASM guidelines, not the entire study protocol.
What is the voxel size of the CBCT scan?
What is the measurement error or variance in the CBCT volumetric calculations? The data on repeated CBCT scans or measurements to assess intra/inter examiner variability were not reported in the results.
After importing into Invivo6, was the orientation of the DICOM files standardized in the coronal plane?
Results:
Due to the lack of significance of any relationship between amount of advancement and volume increase, or volume change and AHI reduction, perhaps subgroup analysis based on either responders vs non-responders, or OSA severity would yield interesting information?
Discussion:
Line 249: the authors state that the minimal cross-sectional area was increased but do not show this in their results.
Due to the wide SD of volume change, did any patient airway volumes decrease in size?
The authors should discuss the likelihood that CBCT is not a reliable measure of airway volume, as a reason for the lack of correlation between AHI and volume change. Based on the data presented, it would seem that CBCT has no useful information to provide for the treatment of OSA patients with oral appliances.
Please comment on how the average measurement error of 3.4cm3 between two CBCT of the same patient reported by Ryan et al AJODO 2019 Jul;156(1):53-60 compares to the results reported in this paper.
Conclusion:
Based on the results of no correlation between AHI and volume increase it seems illogical to have your concluding statement be "As a result of their mechanism of action, these devices are able to increase the volume of the upper airway, thereby lessening the severity of the disorder"
Rather a statement that actually reflects the data shown, such as "CBCT measured volume change of the upper airway does not correlate with reduction in AHI in patients treated with oral appliances" would be preferred.
Reviewer 2 Report
This is an interesting polysomnographic study evaluating upper airways volume in patients who underwent mandibular advancement device therapy. Topic itself is not novel, and entire newly formed dental sleep medicine branch is mostly focused around these, however method of evaluation might be of potential interest for some readers. Hence, some crticism should be raised:
Abstract:
Background: Authors started with aim or hypothesis instead of brief OSAS introduction - please reformulate
Conclusions aren't novel, it is very well known so far. Try reformulating these according to your own achieved study results. See below.
Keywords: I suggest adding a few more according to MeSH - this will improve manuscript availability through search engines and boost future citations
Introduction:
L34 - there is no mention about screening tools prior to PSG like questionnaires, Mallampati score, etc. please incorporate and cite https://www.mdpi.com/2076-3417/11/9/3764
L47 - SDB abbreviation requires name in full, when authors firstly use it, as there is none
L60-L64 - Authors wrote here (with relevant citations) that there is a solid evidence in terms of MADs efficacy - hence an entire aim and conclusions in Abstract section should be re-written accordingly
Materials and methods
L85 - provide a relevant number. Why the study was not randomized is already written in limitations. Please remove word 'non-randomized' from here and rather focus on elaborating its' being longitudinal and the way it was blinded.
L89-98 - please clarify inclusion and exclusion criteria
L100-103 - pleae provide timeframe for both CBCT scan acquisitions
L119 - George gauge is missing the manufacturers' data
L126 - Invivo6 app is missing the manufacturers' data
L143-144 - Was AHI not standardized? Please elaborate briefly why exactly use 5 and less, with relevant (up to date) citations as these are contradictory indeed, which authors mentioned. Also, related reference 18 is dated back into 2018 - many more papers were published recently. These informations are located later on within Discussion section, but should be put in here for clarity
Statistical methods are valid and up for the task
Results
It would be good to provide a CONSORT diagram here, not just plain descriptive analysis
Discussion
Study limiations may require its' own sub-heading
Round 2
Reviewer 2 Report
All of my remarks were addressed successfully. I am satisfied now. Therefore I believe the manuscript is ready for publication.